# A Narrative Review of the Association between Prematurity and Attention-Deficit/Hyperactivity Disorder and Accompanying Inequities across the Life-Course

**DOI:** 10.3390/children10101637

**Published:** 2023-09-30

**Authors:** Yarden S. Fraiman, Genevieve Guyol, Dolores Acevedo-Garcia, Andrew F. Beck, Heather Burris, Tumaini R. Coker, Henning Tiemeier

**Affiliations:** 1Beth Israel Deaconess Medical Center, Harvard Medical School, Boston, MA 02215, USA; 2Boston Medical Center, Boston University Chobanian & Avedisian School of Medicine, Boston, MA 02218, USA; 3Heller School of Social Policy and Management, Brandeis University, Waltham, MA 02453, USA; 4Cincinnati Children’s, University of Cincinnati College of Medicine, Cincinnati, OH 45229, USA; 5Children’s Hospital of Philadelphia, Perelman School of Medicine at the University of Pennsylvania, Philadelphia, PA 19104, USA; 6Seattle Children’s, University of Washington School of Medicine, Seattle, WA 98105, USA; 7Harvard T.H. Chan School of Public Health, Boston, MA 02115, USA

**Keywords:** attention deficit/hyperactivity disorder, ADHD, preterm birth, prematurity, life-course

## Abstract

Preterm birth is associated with an increased risk of neurodevelopmental and neurobehavioral impairments including attention-deficit/hyperactivity disorder (ADHD), the most common neurobehavioral disorder of childhood. In this narrative review, we examine the known associations between prematurity and ADHD and highlight the impact of both prematurity and ADHD on multiple domains across the pediatric life-course. We develop a framework for understanding the health services journey of individuals with ADHD to access appropriate services and treatments for ADHD, the “ADHD Care Cascade”. We then discuss the many racial and ethnic inequities that affect the risk of preterm birth as well as the steps along the “ADHD Care Cascade”. By using a life-course approach, we highlight the ways in which inequities are layered over time to magnify the neurodevelopmental impact of preterm birth on the most vulnerable children across the life-course.

## 1. Introduction

In the United States, approximately 10.5% of babies are born premature (birth prior to 37 weeks), equating to roughly 385,000 preterm infants born each year [1]. Lower gestational age at birth is associated with a higher risk of neurologic complications in the neonatal period, such as intraventricular hemorrhage and periventricular leukomalacia. Prematurity is also associated with several other medical comorbidities such as motor impairment, bronchopulmonary dysplasia, necrotizing enterocolitis, and retinopathy of prematurity [2,3,4]. While medical comorbidities associated with preterm birth are well described in the neonatal period, ongoing research has sought to understand the long-term impacts of prematurity across the life-course.

In this narrative review, we explore the impact of prematurity on neurodevelopmental and neurobehavioral outcomes across the life-course, specifically through the lens of attention-deficit/hyperactivity disorder (ADHD), the most prevalent neurobehavioral diagnosis in childhood and adolescence [5]. Furthermore, we highlight the ways in which racial, ethnic, and socioeconomic inequities in preterm birth, pediatric health services utilization, and ADHD diagnosis and treatment are layered. Over time, inequities are potentially magnified through repeated, clustered insults at critical periods (moments during the life-course that, if interrupted, have inalterable effects across the life-course) and sensitive periods (moments during the life-course that have life-long impacts on the life-course).

## 2. Epidemiology of Attention-Deficit/Hyperactivity Disorder (ADHD)

Attention-deficit/hyperactivity disorder is a neurobehavioral condition that typically presents in early childhood with symptoms that include hyperactivity, impulsivity, and/or inattention [5]. ADHD is the most common neuropsychiatric diagnosis of childhood, affecting approximately 10% of children in the U.S. (an estimated 5.4 million children between 6 and 17 years of age) [5]. One large retrospective cohort study of ADHD prevalence using deidentified claims data of privately insured children found that approximately 10% of children were diagnosed with ADHD before age 10 [6]. There is a well-known gender difference in ADHD diagnosis; male children have a higher likelihood of clinically significant symptoms and diagnosis than their female peers [6,7].

The prevalence of ADHD diagnosis and treatment in the U.S. has notably increased over the past two decades [8,9,10]. The etiology of the prevalence increase in ADHD diagnosis is unclear, though some have suggested that increased diagnosis is due to changes in the diagnostic criteria for ADHD in the Diagnostic and Statistical Manual of Mental Disorders (DSM) [7]. However, this likely does not fully account for epidemiologic trends over time.

ADHD is also the most expensive chronic health condition for children and adolescents. In a large study of 2.9 billion patient records from 183 sources of data, Bui et al. found that ADHD results in USD 20.6 billion in healthcare spending in the United States each year [11]. Additionally, Guo and colleagues found that while children with ADHD accounted for only 5.4% of the New York State Medicaid population, they accounted for over 18% of the total costs; the costs of children with a diagnosis of ADHD were an average of 3.2 times greater than those without an ADHD diagnosis [12]. These additional costs are due to behavioral health services and medication costs [11,12].

The management of ADHD begins with symptom recognition, either by family members, teachers, primary healthcare providers, or the youth themselves. After symptom recognition, children often present to a healthcare provider for evaluation and potential diagnosis. Following diagnosis, individuals and families ideally partner with their healthcare team to determine treatment plans which can include medication—most commonly, stimulants, selective norepinephrine-reuptake inhibitors, and alpha-2-adrenergic agonists—behavioral therapy, or observation [5,13,14]. Finally, the maintenance period includes surveillance, monitoring, and ongoing support to ensure appropriate response to treatment [13]. This process of symptom recognition, referral, evaluation, diagnosis, treatment decisions, and maintenance represents a cascade of events which we refer to here as the “ADHD Care Cascade”. (Figure 1) Failure to enter the “ADHD Care Cascade” or to complete any step along the “ADHD Care Cascade” can lead to suboptimal outcomes for individuals with ADHD. For example, if symptoms are not recognized, a diagnosis of ADHD cannot be made and, thus, access to treatments, supports, and interventions is less likely, and, in some cases, may not be possible.

ADHD is a clinical diagnosis; there are no specific cognitive, metabolic, or neurologic markers or tests for ADHD. This presents a potential barrier to uniform detection of symptoms across varying contexts. In fact, some have suggested that ADHD is a cultural construct, defined by cultural, educational, social, and family norms around appropriate childhood behavior [15]. This may explain the differing rates of ADHD prevalence across countries and geographic regions as well as diagnostic variation related to age at school entry [16,17].

### Association and Pathophysiology of Prematurity and Attention-Deficit/Hyperactivity Disorder

The epidemiologic association between preterm birth and ADHD has been documented across multiple studies and populations. A large birth cohort study of over 110,000 children found that preterm infants had a higher risk of ADHD symptoms compared to their term peers [18]. An analysis of sibling-matched preterm- and term-born siblings confirmed the relationship between preterm birth and ADHD, indicating that the measured association is not due to unmeasured genetic and environmental confounding [18]. The risk of ADHD has also been found to be higher with lower gestational age at birth [19,20,21]. Using a population-based birth cohort in Sweden, Beer and colleagues found a negative linear relationship between gestational age and ADHD diagnosis that persisted among medically indicated and spontaneous preterm births [19]. A large Finnish population-based cohort had similar findings of higher risk of ADHD with lower gestational age at birth [22]. Notably, children born very preterm (less than 33 weeks gestational age at birth), have a two to three times higher risk of being diagnosed with ADHD than their term-born peers [20,23]. Among infants born extremely preterm (less than 29 weeks gestational age birth) or extremely low birth weight (ELBW; birth weight less than 1000 g), there is a fourfold higher risk of developing ADHD compared to their term-born peers [20].

Conflicting evidence exists regarding the risk of ADHD among late preterm (gestational age between 34–36 weeks at birth) infants. Some studies have found no association, while others continue to report a higher risk of ADHD among children born late preterm, albeit a lower risk than very or extremely preterm-born children [18,20,22].

In addition to the elevated risk of ADHD among preterm-born children compared to term-born children, there are also several risk factors specific to preterm-born children that further increase the risk of ADHD. For example, while small-for-gestational-age (SGA; birth weight less than 10 percentile) children have a higher risk for ADHD compared to appropriate-for-gestational-age-weight infants, researchers have found that the association between SGA and a diagnosis of ADHD is stronger among preterm-born children [19]. Cochran et al., using the Extremely Low Gestational Age Newborns (ELGAN) Study, a longitudinal study of children born less than 28 weeks of gestation, demonstrated that diabetes and overweight status in the birthing person also contribute to the development of ADHD in extremely preterm-born children. An overweight birthing person had a 55% increased risk of having an ELGAN with ADHD, while obese birthing people had a 65% increased risk. Diabetes also increased the risk of ADHD, and these joint risks had both independent and multiplicative impacts on risk [24]. There are also associations between intraventricular hemorrhage and bronchopulmonary dysplasia among preterm infants and the development of ADHD, though the data are often conflicting [21].

Additionally, other pregnancy risk factors associated with ADHD in term-born children, such as hypertensive disorders of pregnancy and maternal smoking, were not consistently associated with the development of ADHD in preterm-born children [24]. Social risk factors for the development of ADHD among term-born infants are less important among preterm-born children, whose risk factors for the development of ADHD are mostly related to “medical variables” [20]. Furthermore, the epidemiologic gender differences of ADHD are less clear among preterm-born children [20].

There are also unique characteristics of ADHD among preterm-born children, suggesting a unique subtype and pathophysiology in this population. ADHD is divided into three subtypes: hyperactive or impulsive, inattentive, or combined hyperactive and inattentive. While the combined phenotype is most common in the general population, among preterm-born children, the most common ADHD phenotype is the inattentive subtype [18,20]. This may lead to an increase in missed diagnoses because of the atypical presentation.

Some have suggested that the increased risk of ADHD among preterm infants may be due to a systematic bias as a result of increased medical monitoring, testing, and screening that children born preterm receive, such as through Early Intervention and high-risk infant follow-up programs that may systematically screen for ADHD symptoms [25,26,27]. However, given the unique aspects, qualities, risk factors, and characteristics of ADHD among preterm infants, it is unlikely that the increased monitoring of preterm infants explains the excess of ADHD diagnosis among preterm-born children.

## 3. Burden of ADHD in Childhood, Adolescence, and Adulthood

ADHD has a significant and lasting impact across the life-course for children, adolescents, and adults. Most prominently, ADHD has a significant impact on education, schooling, and educational attainment. Children with ADHD are more likely to have a reading, math, or spelling disability. They are also more likely to need to repeat a grade, be expelled, or drop out from school [7]. One study of the 2001 National Health Interview Survey which included over 10 thousand children between ages 4 and 17 found that 94% of parents of children with ADHD reported impairments at school [7]. In adolescence, children with ADHD are less likely to complete high school, and less likely than their peers to enroll in four-year colleges [28].

Beyond schooling, ADHD can affect multiple domains of health-related quality of life for children and adolescents [7]. Klassen et al., in a cross-sectional study of 259 children referred to an ADHD clinic in British Columbia, Canada, found that despite similar physical health, children with ADHD had significantly worse psychosocial health in all measured domains including role/social limitations as a result of emotional–behavioral problems, self-esteem, mental health, general behavior, emotional impact on parents, time impact on parents, family activities, family cohesion, and overall psychosocial health. Furthermore, higher ADHD symptom burden was associated with an increased impact on psychosocial health [29]. In a large population-based study using the National Health Interview Survey, Cuffe et al. found that nearly 75% of children and adolescents with ADHD have impairments at home [7]. Furthermore, children with ADHD have lower scores on measures of family cohesion and family activity [29]. Children with ADHD are also more likely to engage in high-risk social behaviors, as evidenced by a higher risk of injury and death, increased contact with the juvenile justice system, and higher prevalence of ADHD among incarcerated individuals [13,30,31,32,33,34].

Finally, ADHD is also costly for families. One study found that the total financial burden over the course of the development of a child who has ADHD was more than five times greater than children without ADHD [35,36]. Costs associated with ADHD included direct costs due to missing activities and indirect costs due to family members being fired from their jobs, changing responsibilities, and additional incurred childcare costs necessary to help care for children with ADHD [35].

The 2014 National Survey of the Diagnosis and Treatment of ADHD and Tourette Syndrome found that among children 8–17 years of age, 44.3% of families reported one or more negative financial impacts as a result of a child’s ADHD diagnosis. These included reducing work hours and avoiding job changes to ensure adequate health insurance coverage. Over 14% of families reported that a member of the family stopped working due to the needs of the child with ADHD. More severe and earlier diagnosis was associated with a greater financial impact [30]. Notably, the financial impact of ADHD can affect the treatment and management of ADHD in children. One study found that more than 11% of families with a child with ADHD had unmet needs for ADHD treatment [30].

ADHD can also have financial implications on individuals with ADHD in adulthood. Adults with ADHD are more likely to work entry-level jobs or be unemployed [31,32]. They also have lower lifetime earnings and lower rates of financial independence compared to their peers without ADHD [31,33].

It has been noted that the prevalence of ADHD diagnosis decreases with increasing age, though these patterns may not persist for preterm-born children and adults [21,37,38]. While some studies have found adults born preterm do not have increased rates of ADHD self-reported symptoms, large population-based studies have found that adults born preterm have higher rates of ADHD diagnosis compared to their term-born peers. Thus, while self-reported ADHD symptom prevalence may not vary between preterm- and term-born adults, there is a higher prevalence of ADHD diagnosis in preterm-born adults. The etiology of this finding is unclear and may be due to increased contact with the healthcare system for preterm-born adults as a result of additional medical comorbidities or due to a unique ADHD phenotype that differs from ADHD in term-born individuals.

Notably, the impact of ADHD on psychosocial health, educational attainment, and financial well-being has not been specifically studied among the preterm-born population. For example, while prematurity is associated with costs for families and societal costs, it is not known what proportion of the costs associated with prematurity may be due to ADHD [39]. Due to health-related impacts of prematurity and the increased risk of physical and functional impairments among preterm-born children, we hypothesize that the impact of ADHD across the life-course is likely magnified in preterm-born children. However, further study is needed to understand the specific impact of ADHD on the life of preterm-born children and how it may be different from the impact of ADHD on the life of term-born individuals.

## 4. Layered Inequity in Preterm Birth and along the “ADHD Care Cascade”

### 4.1. Neonatal Inequities

When considering the life-course impacts of prematurity, it is important to recognize that there are significant racial and ethnic inequities in preterm birth in the United States. Race is a social construct and health differences across racial and ethnic populations are not reflective of biology, genetics, or individual behavior. Rather, these differences represent the effects of structural, institutional, interpersonal, and internalized racism [40,41,42,43,44]. Racial and ethnic inequities in access to medical care, provision of medical treatment, and health outcomes are recognized phenomena in the U.S. healthcare system affecting birthing people, their infants, and children across their life-course. Black and Hispanic/Latinx (i.e., minoritized), compared to white, birthing people have an increased risk of preterm and low birthweight births, though the Black-white inequity is considerably the most severe [45,46,47,48,49]. Once born, infants born to Black birthing people experience lower-quality neonatal care and have an increased risk of adverse outcomes [49,50,51,52,53,54,55,56]. These include a higher risk for sepsis, necrotizing enterocolitis, intraventricular hemorrhage, and bronchopulmonary dysplasia, all of which are associated with worse neurodevelopmental outcomes for preterm-born children and may contribute to inequities across the pediatric life-course [57,58,59,60,61,62,63].

Following discharge from the Neonatal Intensive Care Unit (NICU), Black and Hispanic/Latinx preterm infants are less likely to receive important follow up in high-risk infant follow-up clinics and to be referred to and receive services from Early Intervention [64,65,66,67,68,69,70,71]. This inequity in preterm health services utilization is important because studies have found that consistent participation in high-risk infant follow-up clinics, as well as participation in Early Intervention, may improve short- and long-term neurodevelopmental outcomes [72,73,74,75,76]. The lack of access to these services may compound the inequity in preterm birth over the life-course.

### 4.2. Inequities in the “ADHD Care Cascade”

There are known racial and ethnic inequities in multiple neurodevelopmental outcomes including ADHD. However, these can be difficult to fully understand because the process of symptom recognition, diagnosis, and treatment along the “ADHD Care Cascade” is complex and represents a myriad of research outcomes to interrogate. Notably, at each step of the “ADHD Care Cascade” where racial and ethnic inequity is identified, it is racism in its many forms that is the root cause. Furthermore, as the “ADHD Care Cascade” is a series of interconnected steps, inequities in one part of the cascade can be propagated, and potentially magnified, at later steps of the “ADHD Care Cascade”. For the purposes of this narrative review, we build on the conceptual framework of Eiraldi et al. and highlight three points along the “ADHD Care Cascade”: (1) symptom recognition, (2) diagnosis, and (3) treatment [77].

### 4.3. Symptom Recognition Inequities

The process of entering the “ADHD Care Cascade” and thus being eligible for diagnosis, management, and treatment is contingent on symptom recognition. While there are no genetic or biologic differences between racial and ethnic populations, symptom recognition is racialized and inequities exist, suggesting that racism, in its multiple forms, contributes to inequitable symptom recognition. In a large nationally representative cohort, Cuffe et al. found that Black and Hispanic/Latinx children had a higher prevalence of parent-reported symptoms on the Strength and Difficulties Questionnaire, a valid measure of ADHD [7,78]. Others have identified similar trends where Black children are more likely to be identified as having symptoms consistent with ADHD in parental-completed Diagnostic Interview Schedule for Children Predictive Scales [79]. It is unclear whether the increased symptom recognition by parents is a result of cultural differences in parental expectations, variation in the interpretation of child behavior, differences in parental education, or differences in child behavior that occur due to family exposure to experiences of structural racism, racial trauma, and stress. Another study found that when observing the same video of a child’s behavior, white teachers reported more ADHD symptoms for Black children than their Black parents did, and white teachers with more negative attitudes toward Black people gave higher ADHD symptom ratings than teachers with less negative racial attitudes [80].

### 4.4. Diagnostic Inequities

Irrespective of the etiology of higher rates of parental and teacher symptom recognition in Black children, multiple studies have identified that minoritized children are less likely to receive a diagnosis of ADHD. Despite lower rates of symptom recognition, parents of white children are more likely to report that they were told by a healthcare provider that their child has ADHD [7,79,81]. In one study of the Early Childhood Longitudinal Study-Birth Cohort (ECLS-B)—a large nationally representative cohort of U.S.-born children—in unadjusted models, Black children were 70% less likely to receive an ADHD diagnosis compared to their white peers. Furthermore, the inequity persisted after adjusting for markers of behavior, suggesting that even when controlling for behavioral phenotypes, Black, as well as Hispanic/Latinx, children are less likely to receive a diagnosis than their white peers [81]. Similarly, other studies have identified that parents of Black and Hispanic/Latinx children were less likely than parents of white children to report having a child with ADHD, and that these inequities persisted even after adjusting for poverty, birth weight, and insurance coverage [82].

The racial and ethnic inequities in ADHD diagnosis begin as early as five years of age and persist across the pediatric life-course into childhood [10,79,83,84]. These inequities in ADHD prevalence persist although the diagnosis rates tend to increase with age [10,85]. Notably, we identified one study that found that being Black was associated with a higher likelihood of ever receiving a diagnosis of ADHD, though this was not the case for Hispanic/Latinx children in the same study [86].

It remains unknown why diagnostic inequities exist for minoritized individuals. To begin, diagnostic tools may not have equal validity among different racial and ethnic populations. For example, one study found that the Diagnostic Interview Schedule for Children had differential validity among Black and white children; when seven items with high variation were removed, the diagnostic inequities decreased [87]. However, inequities in diagnosis are likely not solely attributable to variations in testing efficacy. Instead, variations in ADHD diagnosis across regions, countries, cultures, classroom style, school environment, and socioeconomic factors suggest that forces such as institutional racism, structural racism, interpersonal racism, and bias may contribute to inequities in ADHD diagnosis [15,84]. As Shi et al. explain, “it is possible, for example, that identical behavior displayed by Black and non-Hispanic white children may be interpreted differently based on race-based expectations for the behavior of children, and thus, behavior that is identified as disordered in white children might be inappropriately interpreted as normal in Black children” [6].

The lack of a diagnosis often precludes minoritized children with ADHD symptoms and impairments from accessing medically indicated, evidence-based interventions that have a positive impact on educational, social, and lifetime earnings.

### 4.5. Inequities in Treatment

Once diagnosed with ADHD, there continue to be significant inequities in access to treatments. As noted above, white children are more likely than minoritized children to receive a diagnosis of ADHD. Once diagnosed, white children are also significantly more likely to receive medication than other racial and ethnic groups [82,83,88,89]. Similar to inequities in symptom recognition and diagnosis, these inequities are seen as early as five years of age and persist [6,81]. Analyses by Coker et al. limited to children with ADHD symptoms or ADHD diagnoses found that Black and Hispanic/Latinx children were less likely than their white peers to receive medications for the treatment of ADHD [79]. Furthermore, these inequities persisted at all levels of symptom severity [79]. As Coker and coauthors state, the inequities in diagnosis and treatment are not related to overdiagnosis or overtreatment in white children but, instead, point to underdiagnosis and undertreatment in minoritized children [79]. Notably, there are similar inequities in medication use that negatively affect those living in homes where the primary language is not English and those in lower socioeconomic strata who are also less likely to receive medication treatment for ADHD [82,86]. These inequities may continue to be magnified over time. For example, racial bias is associated with negative perception of disruptive classroom behaviors due to ADHD. Thus, minoritized children with untreated ADHD may be more likely to experience exclusionary discipline which then further negatively affects school engagement and is also associated with juvenile justice system involvement [90].

When considering the etiology of racial and ethnic inequities in treatment, one must consider the potential of provider biases, both implicit and explicit, that may contribute to inequities in treatment allocation [6,91,92,93]. However, inequities are also propagated above and beyond individual-level bias at the institutional and structural level [40,41,42,43,44]. For example, clinic location, hours of operation, and available resources may contribute to inequities in access, while school culture, cultural and racialized behavioral norms, and public school funding may contribute to inequities at the structural and institutional level.

The inequities in ADHD symptom recognition, diagnosis, and treatment allocation have not been specifically studied in the preterm population. However, we believe that the inequities along the “ADHD Care Cascade” are likely magnified for preterm-born children. We believe that this is likely due to added medical and functional comorbidities that preterm-born children experience as a result of their preterm status, compared to term children, which may make it more difficult to navigate the “ADHD Care Cascade”. Furthermore, the preterm-born population is disproportionately minoritized due to inequities in preterm birth. The intersection of prematurity and race with ADHD likely compound the inequities along the “ADHD Care Cascade”. In other words, the underdiagnoses and undertreatment of minoritized children with ADHD compounds previous inequities in preterm birth, further disadvantaging minoritized children over time. However, further study is needed to fully understand how preterm-born minoritized children traverse the “ADHD Care Cascade”.

## 5. Other Considerations: The Impacts of Prematurity and ADHD on School Readiness and Life-Course Implications

School readiness at kindergarten entry is associated with long-term academic, economic, and health outcomes [94]. Children who are school-ready have higher overall educational attainment, increased adult earnings, and lower prevalence of cardiometabolic disease in adulthood [95,96]. School readiness is a multidimensional concept involving both the child’s readiness for school in multiple domains including physical health, social–emotional development, attitudes towards learning, language skills, preacademic knowledge, and the school’s readiness for the child [97].

There are several mechanisms through which prematurity and ADHD each affect the child’s readiness for school. Neurodevelopmental differences between the term and preterm brain are thought to explain associations between decreasing gestational age and lower teacher assessment of school readiness, kindergarten math and reading scores, and executive function at school entry [98,99,100]. Many children born preterm also have conditions such as chronic lung disease and intraventricular hemorrhage which are independently associated with decreased kindergarten readiness and also affect neurodevelopment [101]. Moreover, behaviors typical to ADHD, such as difficulty concentrating on tasks and interacting with peers, affect school readiness [102]. While differences between children with and without ADHD are most pronounced in the areas of attitudes towards learning and social–emotional development, there is a high prevalence of difficulty in multiple domains of school readiness among children with ADHD [103]. Thus, both prematurity and ADHD are independently associated with decreased school readiness.

While prior research has investigated the individual impacts of prematurity and ADHD on kindergarten readiness, less is known about their combined effect. Since school readiness has lifelong consequences, future work should study the extent to which interactions between these two factors affect school readiness and the short- and long-term life-course contribution of school readiness to inequities.

## 6. Discussion

ADHD serves as a case study for the long-term impacts of prematurity on childhood neurodevelopmental outcomes across the life-course. The association between prematurity and the development of ADHD has been well documented in the literature, and there is likely a specific phenotype of ADHD in preterm-born children characterized by more symptoms of inattentiveness. ADHD is an important diagnosis to consider among preterm-born children because it is the most common neuropsychiatric diagnosis of childhood and has a significant impact on childhood learning, social interactions, and school achievement. Additionally, it is associated with long-term health, safety, and financial independence. Importantly, treatments for ADHD that improve outcomes are available.

However, it must be recognized that the risk of preterm birth is racialized, as are neonatal adverse outcomes. In childhood, racial and ethnic inequities continue to persist, as we highlighted, in (1) symptom recognition, (2) diagnosis, and (3) treatment of ADHD along the “ADHD Care Cascade”. Thus, it is important to recognize not only the long-term life-course impacts of prematurity, but also to specifically address racialized long-term outcomes for preterm-born children.

As shown in this narrative review, the repeated and clustered insults of racism, in its multiple forms, which lead to racial and ethnic inequities in the prenatal, perinatal, neonatal, and pediatric period, have significant and compounding impacts on the life-course health and well-being of preterm-born children, including when viewed through the lens of ADHD. Layered onto the impact of premature birth and racial and ethnic health inequity are additional inequities in access, healthcare utilization, economic opportunity, and social capital. Further study must not only investigate the impact of preterm birth on a single outcome, but, instead, consider the life-course impact of preterm birth and the ways in which inequities early in life may change and potentially widen over time. Specifically, researchers must seek to identify strategies to prevent the long-term adverse neurodevelopmental outcomes of preterm-born children while also identifying the modifiable structural and institutional drivers of inequity in neurodevelopmental outcomes over time. The drivers of inequity are repeated and clustered insults across the life-course of minoritized individuals. and along the care cascades they traverse. Thus the effect of these drivers is magnified; widening gaps overtime. As such, creating meaningful interventions to close equity gaps in neurodevelopment are sorely needed and long overdue.

## Figures and Tables

**Figure 1 children-10-01637-f001:**
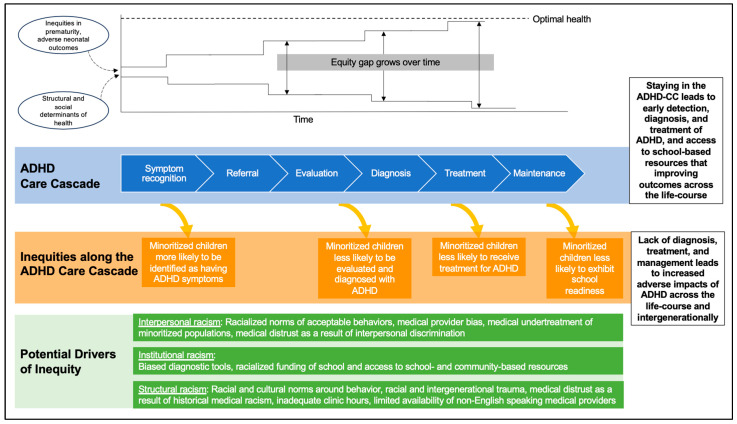
The “ADHD Care Cascade” and the impact of inequities across the life-course.

## Data Availability

Not applicable.

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
