# Peer review of "A Narrative Review of the Association between Prematurity and Attention-Deficit/Hyperactivity Disorder and Accompanying Inequities across the Life-Course"

_children, 2023, doi:10.3390/children10101637_

Round 1
Reviewer 1 Report
Dear authors,
I am glad that I had the opportunity to review this excellent article. Well done.
The life path of all children is notably impacted by various factors, including the level of parents' education, surrounding environment, income and the presence of any illnesses or disorders. In cases where a disorder is present, the aforementioned factors have an even greater impact.
Although it's an excellent work, I have one or two things that you need to elaborate:
1. l.173- 182: All the problems mentioned above can lead to executive function disorders, especially working memory, inhibition, and cognitive flexibility. Therefore, it is crucial to emphasize their significance. Read the work of Adelle Diamond, especially: 1. Diamond A. (2000). Close interrelation of motor development and cognitive development and of the cerebellum and prefrontal cortex. Child Development, 71(1):44-56. doi: 10.1111/1467-8624.00117. PMID: 10836557.
2. Diamond, A. (2013). Executive functions. Annual Review of Psychology, 64, 135-168. doi: 10.1146/annurev-psych-113011-143750.
l.276- 279: You haven't mentioned parents' education. In my opinion it is really important for symptom recognition.
Author Response
Thank you very much for your kind comments and thoughtful feedback regarding our manuscript. Below, please find a point-by-point response to the comments.
- Thank you very much for sharing your comments and sources. This source focuses on the link between disorders of executive functioning, such as ADHD, and physical/functional disorders. While the link between cognitive and physical/functional disorders is important, it falls outside the scope of this focused narrative review.
- Thank you for your comment regarding the impact of parental education on the detection of symptoms. We have edited the text to include incorporate this feedback.
Reviewer 2 Report
The manuscript describes a beautiful synthesis of preterm born, ADHD and inequities with efficiency. The authors must be commended for this. I have few suggestions for improvisation:
1) Introduction: I would probably add some texts on the work (Reviews) that has already been done and the justification for the current narrative review.
2) Introduction: Sentence 51 to 55: I would probably not keep this in the introduction section. This probably can go to the main body/conclusion section.
3) Consider adding little more discussion on low/high birth weight and ADHD under the heading “Association and pathophysiology of preterm and ADHD” (Controversies around genetic basis, catch up growth, pre and postnatal nutrition and socioeconomic status).
4) Consider narrating other pregnancy/postnatal preterm pathologies possibly contributing to altered brain pathology/modelling and development of ADHD. (Maternal characteristics/medications/substances (Authors have discussed few) Intraventricular hemorrhage, periventricular leucomalacia, respiratory morbidity, retinopathy of prematurity, microbiota dysbiosis etc). The readers would benefit if the authors could discuss the pros and cons of preterm ADHD prediction scores.
5) Though this is not a systematic review and methods of literature search is not mandatory, I would still encourage the authors to consider reporting a paragraph on literature search for this narrative review.
6) I would probably trim the discussion part; it sounds like the repetition of what has already been written in the main body. I would probably emphasize more on knowledge gap and future direction.
7) Sentence 209 to 213 needs reference
Author Response
Thank you very much for your kind comments and thoughtful feedback regarding our manuscript. Below, please find a point-by-point response to the comments.
- Thank you for your comment. We reference the existing reviews on ADHD throughout the text.
- Thank you for your comment. The sentence in lines 51-55 is part of a summary of the manuscript and thus we believe it is best suited to the “Introduction” so that authors can understand the overall trajectory of the manuscript.
- Thank you very much for your thoughtful comment regarding low birth weight infants and the association with ADHD. We do note some of the literature on this topic in the section “Association and Pathophysiology of Prematurity and Attention-Deficit/Hyperactivity Disorder.” While this is important and likely closely linked to ADHD in preterm-born children, we chose to limit our discussion to the data on prematurity in order to maintain a focused narrative review. The inclusion of data regarding LBW infants we felt is outside the scope of this manuscript.
- Thank you for your comment. In response, we have added a sentence to address the association between bronchopulmonary dysplasia and intraventricular hemorrhage and ADHD.
- Thank you for this comment. We will defer to the Special Issue editors regarding their stylistic preferences.
- Thank you for your comment. Upon review of the Discussion we found that only the first two paragraphs summarize the manuscript’s key points and the bulk of the section deal with future directions and gaps in research. See Line 412-421. If desired by the Special Issue editors we seek further advice on where additional information might be needed.